# Development and Efficacy Evaluation of a Novel Nano-Emulsion Adjuvant for a Foot-and-Mouth Disease Virus-like Particles Vaccine Based on Squalane

**DOI:** 10.3390/nano12223934

**Published:** 2022-11-08

**Authors:** Xiaoni Shi, Kun Yang, Hetao Song, Zhidong Teng, Yun Zhang, Weihao Ding, Aofei Wang, Shuzhen Tan, Hu Dong, Shiqi Sun, Yonghao Hu, Huichen Guo

**Affiliations:** 1College of Veterinary Medicine, Gansu Agricultural University, Lanzhou 730070, China; 2State Key Laboratory of Veterinary Etiological Biology, College of Veterinary Medicine, Lanzhou University, Lanzhou Veterinary Research Institute, Chinese Academy of Agricultural Sciences, Lanzhou 730000, China; 3School of Chemical Engineering, Lanzhou City University, Lanzhou 730070, China

**Keywords:** nano-emulsion, adjuvants, FMD virus-like particles, squalane, immune responses

## Abstract

The successful development of foot-and-mouth disease virus-like particles (FMD-VLPs) has opened a new direction for researching a novel subunit vaccine for foot-and-mouth disease (FMD). Therefore, it is urgent to develop an adjuvant that is highly effective and safe to facilitate a better immune response to be pair with the FMD-VLP vaccine. In this research, we prepared a new nano-emulsion adjuvant based on squalane (SNA) containing CpG using the pseudo-ternary phase diagram method and the phase transformation method. The SNA consisted of Span85, Tween60, squalane, polyethene glycol-400 (PEG400) and CpG aqueous solution. The average particle diameter of the SNA was about 95 nm, and it exhibited good resistance to centrifugation, thermal stability, and biocompatibility. Then, SNA was emulsified as an adjuvant to prepare foot-and-mouth disease virus-like particles vaccine, *BALB/c* mice and guinea pigs were immunized, and we evaluated the immunization effect. The immunization results in mice showed that the SNA-VLPs vaccine significantly increased specific antibody levels in mice within 4 weeks, including higher levels of IgG1 and IgG2a. In addition, it increased the levels of IFN-γ and IL-1β in the immune serum of mice. Meanwhile, guinea pig-specific and neutralizing antibodies were considerably increased within 4 weeks when SNA was used as an adjuvant, thereby facilitating the proliferation of splenic lymphocytes. More importantly, in guinea pigs immunized with one dose of SNA-VLPs, challenged with FMDV 28 days after immunization, the protection rate can reach 83.3%, which is as high as in the ISA-206 control group. In conclusion, the novel squalane nano-emulsion adjuvant is an effective adjuvant for the FMD-VLPs vaccine, indicating a promising adjuvant for the future development of a novel FMD-VLPs vaccine.

## 1. Introduction

Foot-and-mouth disease (FMD) is a disease that spreads rapidly in even-toed animals [1]. In most countries, animals are immunized with the whole virus inactivated vaccines to control the virus; however, safety is a concern [2]. Adjuvants provide an important way to improve the efficacy of FMD vaccines. Hence, the search for specific and targeted adjuvants combined with protective antigens is a new direction for developing novel FMD vaccines [3].

The development of virus-like particles (VLPs) technology has strongly impacted modern vaccinology. Though morphologically similar to native viral particles, VLPs show higher and safer efficiency in stimulating the immune system because they lack replicable viral genetic material [4]. VLPs can be generated using recombinant DNA technology in various exogenous gene expression systems, including yeast, bacteria, mammalian cells, baculovirus systems, plant cell cultures, or plant organisms. VLPs-based vaccines are not only particularly effective and safe, but also have a low cost and can be produced at scale [5]. Therefore, VLPs are expected to be ideal candidates for vaccine development [6]. Compared to inactivated vaccines, VLPs alone do not induce a sufficient specific immune response; they also require the appropriate adjuvants to enhance their immune response. Compared to inactivated and attenuated vaccines, the FMD virus-like particle vaccine is a new safe and effective vaccine [7]. Therefore, it is necessary to identify a new adjuvant with high efficiency, low cost, and low toxicity for FMD-VLPs.

Emulsion adjuvants are usually formulated from oils, such as mineral oil (e.g., Montanide) and squalene (e.g., MF59) and surfactants. They are available in water-in-oil, oil-in-water and water-in-oil-in-water dispersion forms [8]. The oil emulsion adjuvant has a high antigen adsorption capacity and can bind to different types of antigens [9]. By adding immune boosters, nano-emulsion adjuvants can stimulate both humoral and cellular immunity in the body; therefore, they have become one of the more widely used adjuvants in animal vaccines. Montanide ISA-206, a mineral oil-based adjuvant, is produced by Seppic (Shanghai) Chemical Specialities Co., Ltd. (Shanghai, China)., and is presently used for formulating FMD vaccines in many South American and Asian countries.

Squalane is a saturated aliphatic hydrocarbon with low toxicity, derived from the hydrogenation of squalene, which is found in cod liver oil, rice, olives, and soybeans [10]. Based on its strong stability and biocompatibility, squalane is currently used in many vaccines and drug delivery emulsions [11]. For instance, emulsions MF59 (Novartis, Basel, Switzerland), AS03 (GlaxoSmithKline, Brentford, UK), and AF03 (Sanofi, Paris, France) are squalene-based and have been used as adjuvants in anti-influenza virus vaccines [12]. Whether squalane can act as an effective adjuvant to enhance the immune response in an FMD-VLPs vaccine remains unclear.

Compared with conventional emulsions, nano-emulsion has a low viscosity, small particle size, good stability and fewer toxic side effects [13]. As a new type of drug carrier, nano-emulsion has many advantages that are incomparable to other drug carriers. Due to these characteristics, they show attractive prospects for development in the field of biologics.

In this study, a basic formulation of a squalane nano-emulsion adjuvant (SNA) was developed by a pseudo-ternary phase diagram, and CpG was added as an immune booster to the water phase. The physicochemical properties of the adjuvant were tested. Furthermore, SNA was emulsified with FMD VLPs, and animal experiments were performed to evaluate the immune response of these VLPs.

## 2. Materials and Methods

### 2.1. Selection of the Surfactant and Cosurfactant

Based on preliminary experiments, we chose squalane (Acmec, Shanghai, China) as the oil phase, Span85 (Aladdin, Shanghai, China) and Tween60 (Sigma-Aldrich, Saint Louis, MO, USA) as the surfactant, polyethene glycol-400 (PEG-400, Acmec, Shanghai, China) as the cosurfactant, and deionized water as the aqueous phase. We used 0.8 g of PEG-400 and 1.2 g of the Span85 and Tween60. In the reaction system, the surfactant and co-surfactant were blended with the oil phase at ratios of 1:9, 2:8, 3:7, 4:6, 5:5, 6:4, 7:3, 8:2, and 9:1. In the surfactant, Span85 and Tween 60 were blended in the ratios of 1:1, 1:2, and 2:1; the appropriate ratio of Span85 to Tween60 was chosen by recording the maximum area of the pseudo-ternary phase diagram [14]. Origin 6.0 software (Origin Lab, Northampton, MA, USA) was used to plot a pseudo-ternary phase diagram, compare the size of the nano-emulsion area, and select the optimal ratio.

After determining the mass ratio of the two surfactants, the mass ratio of the surfactant to the co-surfactant (Km) was further determined. The surfactant (Span85 and Tween60) was coupled with PEG-400 in three groups under different fixed mass ratios of Km (1:1, 2:1, and 1:2), and the total quantity of the mixture (the surfactant and co-surfactant) was maintained at 2.0 g. For each group, squalane was added and mixed well with the surfactant and cosurfactant. The mass ratio of the surfactant and cosurfactant to oil ranged from 1:9, 2:8, 3:7, 4:6, 5:5, 6:4, 7:3, 8:2, to 9:1. The phase diagrams under different Km were developed.

Selection of Span85 and Tween60 ratios, and selection of surfactant and co-surfactant ratios information is provided in the Appendix A.

### 2.2. Preparation of the SNA

From the pseudoternary-diagram, the precise composition was selected for the nano-emulsion. Briefly, we made a mixture by weighing a certain mass of Span85, Tween60, PEG-400, and squalane in proportion to each other. The aqueous phase (with the addition of CpG) was added to the mixture while stirring at 1000 rpm at room temperature until a clear and transparent emulsion formed; then, stirring was continued to ensure that the nano-emulsion was stable.

### 2.3. Characterization of the SNA

The ultrastructure and morphology of the SNA were observed by transmission electron microscope (TEM; HT7700, Hitachi, Tokyo, Japan). The SNA for analysis was diluted 100 times by water and magnetically stirred well; then, 10 μL of drops were placed on a carbon copper grid (300 mesh; Pelco, CA, USA). Next, the samples were allowed to stand at room temperature for 5 min; then, we added 10 µL of 1% phosphotungstic acid (pH 7.4, solarbio, Beijing, China) solution, let it dry naturally and observed it. The average size and zeta potential were measured by dynamic light scattering (DLS) with a Zetasizer-Nano (Malvern Zetasizer Nano ZS90; Worcestershire, UK).

### 2.4. Stability Assessment of the SNA

#### 2.4.1. High-Speed Centrifuge Stability and Thermodynamic Stability

Thermodynamic stability was tested according to the methods described previously [15,16]. Briefly, the SNA prepared freshly were centrifuged at 13,000× *g* for 30 min at 25 °C and 4 °C, maintain for 48 h. Six cycles of centrifugation were performed. After centrifugation, we observed whether phase separation and precipitation occurred.

#### 2.4.2. Long-Term Stability Test

The SNA was stored at 25 °C for 12 months in the dark. Samples were taken at 0, 3, 6, 9 and 12 months to observe the stratification, precipitation and turbidity. We measured the average particle size, zeta potential, PDI and pH, and the microscopic morphology was observed by transmission electron microscopy.

### 2.5. Biocompatibility Evaluation of the SNA

#### 2.5.1. Tissue Toxicity

*Balb/c* female mice were purchased from the Experimental Animal Center of Lanzhou Veterinary Research Institute (Lanzhou, China) (age = 8 weeks; weight = 15–20 g). Two experimental groups were designed in this study: the PBS control group (each mouse was injected with 100 μg sterile PBS) and the SNA experimental group (each mouse was injected with 100 μg SNA) (n = 8 in each group). We observed whether the injection site was red and swollen, and recorded the weekly weight gain of the mice. Approximately 28 days later, the mice were anaesthetized with ether. Then, the heart, liver, spleen, lung and kidney tissues were removed, preserved in 4% paraformaldehyde, embedded in paraffin, and stained with H&E.

#### 2.5.2. Cytotoxicity

PK-15 (Porcine kidney) cells were inoculated into 96-well cell culture plates. Approximately 24 h later, different concentrations of SNA (0 μg/mL, 25 μg/mL, 50 μg/mL, 100 μg/mL, 200 μg/mL, 400 μg/mL, and 800 μg/mL) were added to the cells at 10 μL per well, which were incubated for 24 h. Then, 10 μL MTS (Promega, Madison, WI, USA) reagent was added to each well and incubated for 4 h. The absorbance was measured by an enzyme marker. The cell viability was calculated with the following equation:$$\;\mathrm{Cell}\;\mathrm{survival}\;(\%)=\frac{\mathrm{absorbance}\;\mathrm{value}\;\mathrm{of}\;\mathrm{treatment}\;\mathrm{group}}{\mathrm{absorbance}\;\mathrm{value}\;\mathrm{of}\;\mathrm{control}\;\mathrm{group}\;}\times 100\%$$

### 2.6. Preparation of the FMD-VLP Vaccine with SNA

The expression, purification and assembly of FMD-VLPs in the *Escherichia coli* system have been described in our previous studies [7,17]. Expression, purification, and assembly of FMD-VLPs can be found in the Appendix A. Initially, the mass ratios of adjuvant to antigen phase (FMD-VLP protein solution) was selected as 1:1, 2:3, 1:2, and 3:2 and emulsified by a shear machine at room temperature. After one week, the non-stratified one was selected as having the best emulsification ratio. After several experiments, we finally determined the mass ratio of the adjuvant-to-antigen phase to be 1:1.

### 2.7. Immunization Studies in BALB/c Female Mice

*BALB/c* female mice (6–8 weeks old, 15–20 g) were maintained in a specific pathogen-free (SPF) laboratory and separated into four groups, each including five animals: Group1 was immunized with sterile phosphate-buffered saline (PBS); Group2 was immunized with the 50 μg FMD VLPs; Group3 was immunized with the vaccine containing 50 μg of FMD VLPs and emulsified with the same volume of SNA; Group4 was immunized with a vaccine emulsified with ISA206 adjuvant (containing 50 µg of FMD VLPs). All mice were immunized by intramuscular injection, and serum was collected 7, 14, 21 and 28 days after immunization. The specific antibody levels were detected by using a liquid phase blocking ELISA kit (Lanzhou Veterinary Research Biotechnology Co, Lanzhou, China).

Indirect ELISA evaluated the levels of the specific antibodies, IgG1 and IgG2a in serum. Briefly, microtiter plates (Coster, Corning, NY, USA) were coated with FMDV VLPs (2 μg/mL) in coating buffer (0.05 M CBS, pH 9.6) at 4 °C overnight and then blocked with BSA (1%, m). Then, the plates were washed with PBST (10 mM PBS containing 0.05% Tween 20, pH 7.4) and dried for the subsequent procedure. For determination of IgG1 and IgG2a sera were diluted 1/200 and incubated at 37 °C for 1 h; goat anti-mouse IgG1 andIgG2a (Sigma-Aldrich, Saint Louis, MO, USA) were diluted 1/1000 and incubated at 37 °C for 60 min; HRP-conjugated rabbit anti goat IgG (Sigma-Aldrich, Saint Louis, MO, USA) was diluted 1/5000 and incubated at RT for 30 min; the enzyme substrate 3,3′,5,5′-tetramethylbenzidine (TMB, Surmodics IVD Inc., MN, USA) was added as described by the manufacturer. Then, the absorbance was measured at 450 nm with a microplate reader (Bio-Tek, Winooski, VT, USA) after the reaction was stopped with the stop buffer of sulfuric acid.

The cytokine levels of the IL-1β and IFN-γ in the serum were detected by Quantikine^®^ ELISA kits (R&D Systems, Inc., Minneapolis, MN, USA).

### 2.8. Guinea Pig Immunization with FMD-VLP Vaccine with SNA

#### 2.8.1. Animal Vaccination

Twenty-four guinea pigs, 250–300 g each, were purchased from the laboratory animal centre of Lanzhou Veterinary Laboratory Experimental, China, which were randomly divided into three groups, each containing eight guinea pigs: Group A, SNA-VLPs (FMD-VLPs emulsified with SNA); Group B, ISA206-VLPs (FMD-VLPs emulsified with ISA206); and Group C, phosphate buffered saline (PBS, pH 7.4). Each group received 50 μg of FMD-VLPs (except the PBS group).

#### 2.8.2. Detection of Specific Antibodies and Cytokine Levels

Specific antibody titres of immunized guinea pigs were determined by indirect ELISA as previously described [7]. Briefly, O-type inactivated FMDV was diluted with a coating solution (0.05 M bicarbonate buffer, pH 9.6), and 100 µL was added to each well of a 96-well plate overnight at 4 °C. Then, 100 μL of PBS containing 5% BSA was added to each well for 1 h at 37 °C to be blocked, washed and drained. One-hundred fold dilutions of tested sera were added to the 96-well plate and were incubated at 37 °C for 1 h. Afterwards, the sample sera were removed and washed. Horseradish peroxidase (HRP)-conjugated anti-guinea pig antibody (1:2000) (Sigma, St. Louis, MO, USA) was added and incubated at 37 °C for 1 h. Then, 50 μL of the enzyme substrate o-phenylenediamine (OPD, Sigma, St. Louis, MO, USA) in sodium citrate was added to each well and incubated for 15 min at room temperature. Finally, the reaction was stopped with 50 μL of 2 M H_2_SO_4_ and the OD value was read at 492 nm. Antibody reactivity was reported as OD values.

The cytokine secretion levels of the IL-1β and IFN-γ were measured with an ELISA kit (Shanghai MLBIO Biotechnology Co. Ltd, Shanghai, China), according to the manufacturer’s instructions.

#### 2.8.3. Detection of the Neutralizing Antibodies

The guinea pig serum to be tested was inactivated at 56 °C for 30 min, with eight gradients of fold dilution, starting at 1:4 in 96-well cell culture plates, with two replicates for each dilution. We added 100 TCID50 of the FMDV type O strain O/China99 to each well of diluted serum and incubate for 1 h at 37 °C. A positive control, negative control, cell control, and virus regression control were also designed. Then, 100 µL of a Baby Hamster Syrian Kidney (BHK-21) cells mixture was added to each well and incubated in a CO_2_ incubator for 72 h. Finally, cell lesions were observed under an inverted microscope. The Reed–Munch method calculated the highest dilution of the serum that protected 50% of the cells from cytopathic lesions. The dilution reflected the potency of the serum to neutralize the antibody.

#### 2.8.4. T-Cell Proliferation Assay

The lymphocyte proliferation assay was performed four weeks after immunization, as previously described [18].

#### 2.8.5. Challenge Protocols

The challenge protocols in guinea pigs were performed 28 days after immunization; all guinea pigs were subcutaneously and intradermally challenged with 0.2 mL 100 ID50 of live virus (FMDV/O/China99) on the left back sole. After the attack, the guinea pigs were kept in isolation and observed for more than 7 days. If the guinea pig had no lesions on either hind foot, the set-up was indicated as “protected”. If lesions were present on both hind feet, it was indicated as “no protected” [19].
$$\;\mathrm{Protection}\;\mathrm{rate}\;(\%)=\frac{\mathrm{Full}\;\mathrm{protection}\;\mathrm{guinea}\;\mathrm{pigs}}{\mathrm{Total}\;\mathrm{number}\;\mathrm{of}\;\mathrm{guinea}\;\mathrm{pigs}\;}\times 100\%$$

### 2.9. Statistical Analysis

We analysed all data using SPSS 22.0. The *t*-test was used to determine the difference between the two sample groups. Differences at 0.01 < *p* ≤ 0.05 were considered statistically significant, whereas those at *p* ≤ 0.01 were considered highly significant.

## 3. Results

### 3.1. Preparation of the SNA

In the pseudo-ternary phase diagram, the first axis represents the oil phase, the second axis represents the aqueous phase, and the last axis represents the mixture of surfactant and co-surfactant. The size of the enclosed space formed by the three components is related to the ability to form nano-emulsions, with larger areas indicating greater ability to form nano-emulsions. According to Figure 1, the size of the area for the Smix (Smix represents the ratio of Span85 and Tween60) = 2:1 (Figure 1c) was larger than those for the Smix = 1:1 (Figure 1a) and Smix = 1:2 (Figure 1b). Based on the largest black area in the phase diagram (Figure 1d), Smix = 2:1 was accepted and applied for the preparation of nano-emulsions.

The pseudo-ternary phase diagrams for different Km (Km represents the ratio of surfactants and co-surfactants) are shown in Figure 2. The area for Km = 3:2 (Figure 2c) was larger than those for Km = 1:1 (Figure 2a) and Km = 2:1 (Figure 2b). Based on the largest black area in the phase diagram (Figure 2d), Km = 3:2 was accepted and applied for the formation of the nano-emulsions.

### 3.2. Characterization of the SNA

The nano-emulsions observed by transmission electron microscopy were uniformly sized circles (Figure 3a), with a particle size of about 100 nm. The morphology of FMD-VLPs is shown in Figure 3c. The SNA prepared by the phase conversion method is shown in Figure 3d, and the FMDV-VLP vaccine emulsified with SNA is shown in Figure 3e.

The average particle size of SNA was 95 nm; about 75% of the nano-emulsion particles were concentrated between 68 nm and 105 nm, with a very narrow particle size distribution (Figure 3b), which were consistent with the results obtained from TEM. In addition, we determined the polydispersity index (PDI) of nano-emulsion SNA with a value of 0.4; this PDI value indicates that dispersion is within acceptable limits, and the average potential of the nano-emulsion was −25.76 mV.

### 3.3. Stability of the SNA

Long-term stability is an essential basis for product preservation conditions and validity. Figure 4a–f show the transmission electron micrographs of the SNA with preservation periods of 0, 90, 180, 270, and 360 days, respectively; Figure 4f–h and Figure 4i show the particle size zeta potential, PDI, and pH for different preservation days, respectively. These results showed that the SNA prepared in this experiment did not change significantly in appearance properties when left at 25 °C for 360 d in the dark. This emulsion was stable at room temperature.

The high-speed centrifugation and thermodynamic stability test confirmed that the nano-emulsion was as transparent and homogeneous as the fresh nano-emulsion, with no turbidity and no precipitation appearing (Figure 4j).

### 3.4. Safety Evaluation of the SNA

Within 48 h after injection, the mice took food and water as normal, and there was no redness or swelling at the injection site. Furthermore, the weight gain trends of the mice in different injection groups were similar, indicating that the injection of SNA had no significant effect on the weight gain of the mice (Figure 5a).

Twenty-eight days after injection, the pathological changes of the tissue organs were analysed. Figure 5c shows that no tissue structure injuries or neutrophil infiltration occurred in the SNA group compared to the PBS-immunized group. This indicates that the SNA injection into mice did not cause significant pathological histological toxicity. The PK-15 cells were incubated with different dilution concentrations of SNA, and the results are shown in Figure 5b. The survival rate of the PK-15 cells all remained high when the SNA dilution concentration was as high as 400 μg/mL, which indicates that the SNA had little cytotoxicity in the PK-15 cells (Figure 5b). All the results confirmed that the prepared SNA had good biocompatibility.

### 3.5. Effects of the FMDV-VLP Vaccine with SNA on the Antibody Production in BALB/c Mice

After several experiments, we finally determined the mass ratio of the adjuvant to antigen phase to be 1:1; based on this ratio, we prepared the SNA for the FMD-VLP vaccine and immunized the animals.

The specific antibody test results showed that the SNA-adjuvant vaccine produced higher antibodies than the VLPs and PBS groups at weeks 2–4, comparable to those produced by the ISA206-adjuvant group. This result indicates that the combination of SNA and FMDV-VLPs can significantly enhance the immune response in *BALB/c* mice as compared to VLPs alone (Figure 6a).

The results showed that, at a serum dilution of 1:100, the SNA vaccine group produced a relatively high IgG1, comparable to the ISA206 group, and produced a higher IgG2a than the VLPs and PBS groups and slightly lower than the ISA206 group (Figure 6c). The IgG2a antibody levels reflect the Th1 immune responses, and the IgG1 antibody levels reflect the Th2 immunity. This result also indicates that the SNA-adjuvant vaccination mainly induced humoral immunity in the body but also induced different levels of cellular immune responses.

In the present mouse immunization experiment, the SNA vaccine group produced higher levels of IFN-γ than the ISA206 vaccine group and the VLPs antigen group (Figure 6b). This result again demonstrates that the SNA-adjuvant vaccine induced a Th1-type immune response. Meanwhile, the serum IL-1β in the SNA adjuvant immunization group was higher than in the ISA206-adjuvant group and the VLPs antigen group (Figure 6d).

### 3.6. Evaluation of Immunization Effect after Immunization of Guinea Pigs

As shown in Figure 7 and Table 1, the specific and neutralizing antibody titres were significantly higher in the SNA-VLPs immunized guinea pigs compared to the PBS group, and T lymphocyte proliferation was also promoted. However, compared to the ISA206-VLPs group, there were no significant differences in the specific antibody titres, T lymphocyte proliferation levels and protection rates in the SNA immunization group.

To confirm whether the SNA induced cellular immunity, we measured the expression levels of the related cytokines IFN-γ. The result indicated that the levels of IFN-γ (Figure 7d) were upregulated compared with the PBS group; the IFN-γ levels in the SNA-VLP group were comparable to those in the ISA206-VLP group, and there was no significant difference.

## 4. Discussion

Vaccination is the most effective and economical method with which to control and prevent FMD. FMD whole-virus-inactivated vaccines have been widely used in various countries, but there are inherent risks. The successful development of virus-like particles (VLPs) has opened a new direction for modern vaccines. Recombinant VLPs are a good alternative to traditional vaccines for FMD because they are non-infectious, require low production conditions, and can also be modified to improve their stability [19].

Nanomaterials with targeted delivery and high bioavailability hold promise as a new delivery system for biocontrol, disease prevention, and chemical antimicrobials [20,21]. Compared with other conventional emulsions, nano-emulsions have a particle size of 20–200 nm, good stability, easy storage, easy penetration of tissue barriers, easier uptake by dendritic cells, and fewer toxic side effects [22,23]; so, it is important to apply nano-emulsions as adjuvants in the research and development of vaccines. Joyappa et al. immunized mice and guinea pigs with a foot-and-mouth disease P1-3CD DNA vaccine loaded onto calcium phosphate nanoparticles and induced significant cellular and humoral immunity while effectively protecting immunized mice and guinea pigs from attack by foot-and-mouth disease virus [24]. Huang et al. developed a novel nano-emulsion PELC and immunized mice by combining PELC with inactivated influenza virus vaccine and found that the PELC significantly enhanced the proliferative activity of T lymphocytes, increased the secretion levels of IFN-γ and IL-4, and induced higher levels of specific antibodies [25].

Squalene oil-based adjuvants, such as MF59 and AS03, are components in commercially available influenza vaccines in Europe, with average droplet sizes in the subnanometre range of 155–160 nm [26]. In this study, we successfully prepared the squalane nano-emulsion adjuvant, SNA, with an average particle size of 95 nm, which was smaller than that of the MF59 and AS03 adjuvants. We also used it for the first time as an adjuvant for the FMD-VLP vaccine and conducted immunization experiments in mice and guinea pigs.

The key to the preparation of nano-emulsions is the selection of a suitable oil, surfactant, and co-surfactant, as well as the appropriate ratio between the oil and the mixed surfactant (surfactant and co-surfactant). It has been noted that water/oil nano-emulsions formulated with non-ionic surfactants have no phase separation over a more extended period time, are more stable, and have greater biological efficacy [27]. Therefore, in this study, we chose Span85 and Tween60 as surfactants and determined the exact ratios between the two surfactants and between the surfactant and the co-surfactant using pseudo-ternary phase diagrams. We used a simple low-energy emulsification method (phase change method) to prepare the nano-emulsions by adding the aqueous phase (antigen phase) dropwise to a mixture of oil, surfactant, and co-surfactant and gently stirring until a clear clarified emulsion appears. Therefore, this method is easy and has great advantages for large-scale production.

The stability of the oil–emulsion adjuvant is an essential indicator of the excellent adjuvant properties; we chose the high-speed centrifugation method [28], the thermodynamic stability method, and the long-term storage method to assess the stability of the SNA comprehensively. The results of all three methods proved that the SNA adjuvant had good stability with uniform particles, good dispersion, and no significant changes in zeta potential and particle size after 12 months of storage.

The safety evaluation of vaccine adjuvants is a very important link before clinical application. At present, there are not many normative documents for the safety evaluation of adjuvants in the world, and no unified evaluation standard has been formed. Currently, the safety evaluation of vaccine adjuvants mainly refers to the evaluation method of clinical drugs. In this study, the safety of the SNA was evaluated from two aspects: a cytotoxicity test in vitro and a tissue toxicity test in vivo. The cytotoxicity test showed that the SNA did not affect the growth of the PK-15 cells at medium and low concentrations. In the mouse experiment, the growth condition of mice was not affected after the nano-emulsion injection, and no pathological changes were found in the tissue sections of essential organs. Thus, these data indicate that the SNA is safe as an adjuvant for injection.

CpG has a good safety profile, enhances the antigen-presenting function of DCs, monocytes and macrophages [29], induces B-cell proliferation, indirectly stimulates the immune activity of NK cells, and significantly favours the immune response to the Th1 type [30,31]. Hence, we added CpG to this new nano-emulsion to obtain a better immune effect. It has been documented that the immunization of guinea pigs with CpG as a vaccine adjuvant encapsulated in chitosan-coated poly (lactic acid)-glycolic acid nanoparticles provided ideal protection [32]. It has also been reported that the addition of CpG to the influenza virus VLP vaccine offers complete protection against the 1918 influenza virus [33].

In mice immune experiments, we determined that the SNA adjuvant with the added CpG component increased the serum levels of IgG2a and IFN-γ, with a significant tendency to induce Th1 cell immune responses and was comparable to ISA206 in promoting IL-1β secretion, lymphocyte proliferation and attack protection, showing the same immune-enhancing effect as the positive control ISA206. The immunization and challenge experiments in guinea pigs showed that SNA as an adjuvant for the FMD-VLPs vaccine induced a stronger specific immune response against FMDV. In particular, there was no significant difference in antibody levels and protection between the SNA-VLPs-immunized and ISA206-VLPs-immunized groups. Pervaiz et al. evaluated the adjuvant role of three oil adjuvants (GAHOL, Montanide ISA-206 and ISA-201) in FMD inactivated vaccines. Among them, GAHOL is a home-made oil adjuvant. Thirty days after vaccination, 100% (6/6) of cattle immunized with the Montanide-201 adjuvanted vaccine were protected in a homologous FMD challenge, which was superior to cattle vaccinated with the ISA-206 (66.6%, 4/6) or GAHOL adjuvanted vaccine (50%, 3/6) [34]. In conclusion, compared with the results of previous studies, the SNA adjuvant prepared in the present study with FMD VLPs produced a protection rate in guinea pigs within the ideal range.

## 5. Conclusions

The present study indicated that the SNA, a novel nano-emulsion adjuvant of squalene, is simple to prepare and easy to produce on a large scale, has good biocompatibility and a relatively comprehensive immune-enhancing effect, and can be an effective adjuvant for FMD VLPs vaccines. Therefore, the current study indicates that SNA may become a new conventional adjuvant with good application prospects for FMD therapy. However, whether SNA is suitable for other subunit vaccines and its underlying immune mechanism need further investigation, which is our future work.

## Figures and Tables

**Figure 1 nanomaterials-12-03934-f001:**
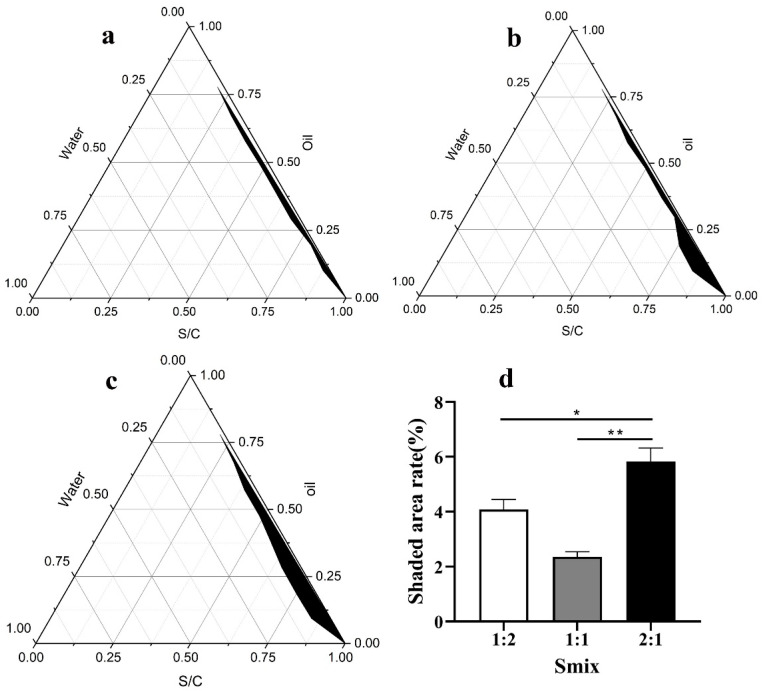
Pseudo-ternary phase diagram obtained with different ratios of Span85 and Tween60. (**a**) Span85:Tween60 = 1:1. (**b**) Span85:Tween60 = 1:2. (**c**) Span85:Tween60 = 2:1. (**d**) The ratio of the shaded area to the area of the ternary phase diagram for the different Smix values (based on the (**a**–**c**)). * *p* < 0.05, ** *p* < 0.01. Notes: S/C represents the total mass of the surfactant and co-surfactant. Smix represents the relative ratio of Span85 and Tween60. Span85 and Tween60 form a surfactant complex.

**Figure 2 nanomaterials-12-03934-f002:**
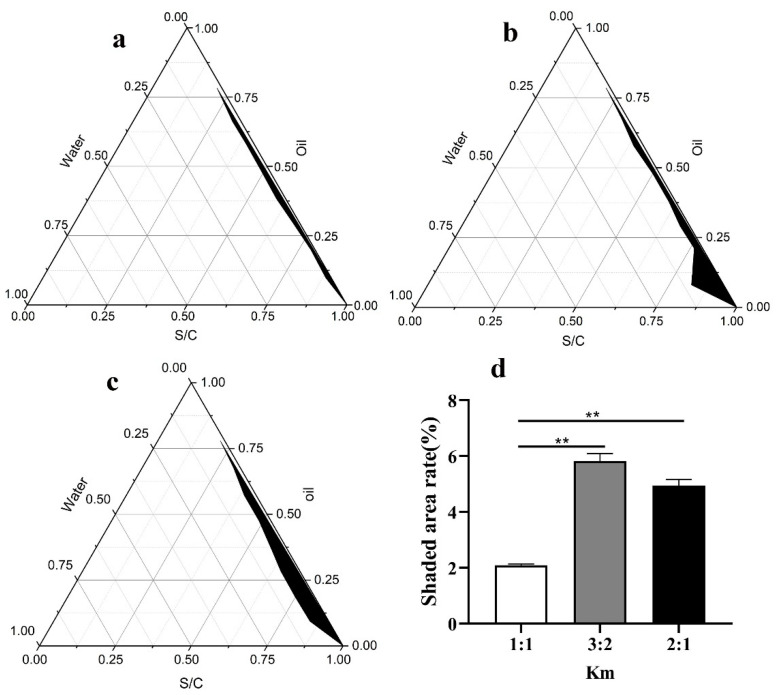
Pseudo-ternary phase diagram for different Km values (**a**) Km = 1:1. (**b**) Km = 2:1. (**c**) Km = 3:2. (**d**) The ratio of the shaded area to the area of the ternary phase diagram for the different Km values. (based on the (**a**–**c**)). ** *p* < 0.01. Notes: Km indicates the relative ratio of the surfactant (Span85 and Tween60) and the co-surfactant (PEG-400).

**Figure 3 nanomaterials-12-03934-f003:**
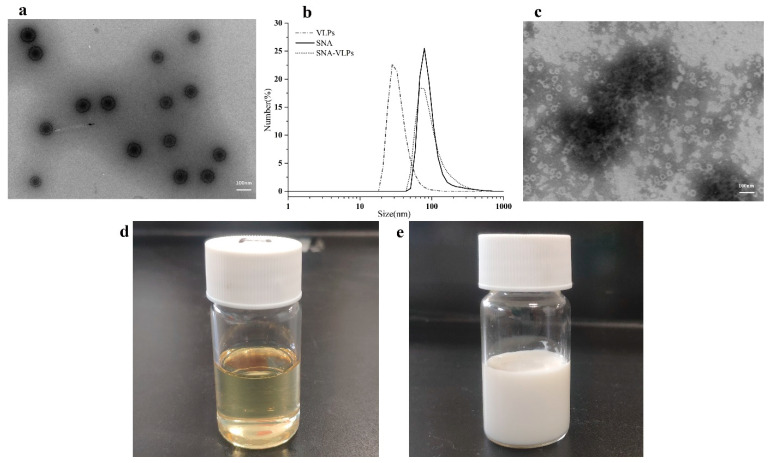
Basic characteristics of the SNA adjuvant. (**a**) The TEM images of the freshly formulated SNA, (**b**) The particle size of the SNA, SNA-VLPs, and FMD-VLPs were measured by DLS. (**c**) The TEM images of the FMD-VLPs. (**d**) A new preparation of SNA. (**e**) A new preparation of the FMD-VLP-SNA vaccine.

**Figure 4 nanomaterials-12-03934-f004:**
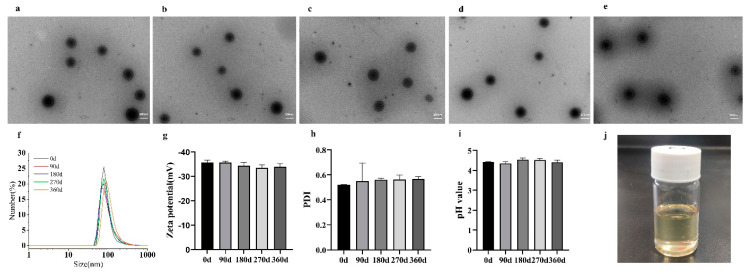
Stability testing of the SNA adjuvants. (**a**) SNA transmission electron micrograph of 0 days. (**b**) SNA transmission electron micrograph of 90 days. (**c**) SNA transmission electron micrograph of 180 days. (**d**) SNA transmission electron micrograph of 270 days. (**e**) SNA transmission electron micrograph of 360 days. (**f**) The particle size for different preservation periods. (**g**) The zeta potential for different preservation periods. (**h**) The PDI for different preservation periods. (**i**) The pH for different preservation periods. (**j**) The SNA after the high−speed centrifugal and thermodynamic stability tests.

**Figure 5 nanomaterials-12-03934-f005:**
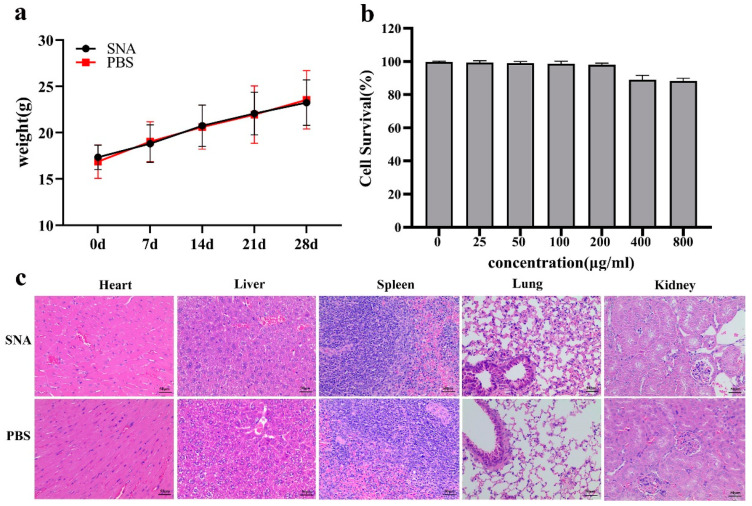
Safety evaluation of the SNA adjuvants. (**a**) The body weight of the immunized mice at 0, 7, 14, 21, and 28 days after immunization. (**b**) The cytotoxicity of different concentration of SNA on the PK-15 cells. (**c**) Histopathological changes in the heart, liver, spleen, lung and kidney of mice after intramuscular injection of SNA and PBS.

**Figure 6 nanomaterials-12-03934-f006:**
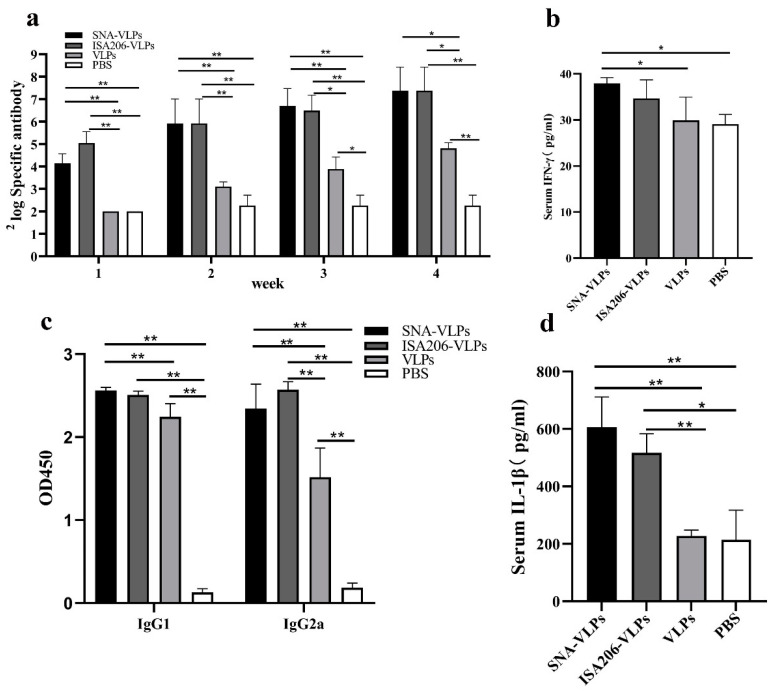
Results of the immunization experiments in mice. (**a**) Specific antibody levels in the mice immunized with various vaccines. (**b**) IFN-γ levels in serum. (**c**) Levels of IgG1 and IgG2a. (**d**) IL-1β levels in serum. * *p* < 0.05, ** *p* < 0.01.

**Figure 7 nanomaterials-12-03934-f007:**
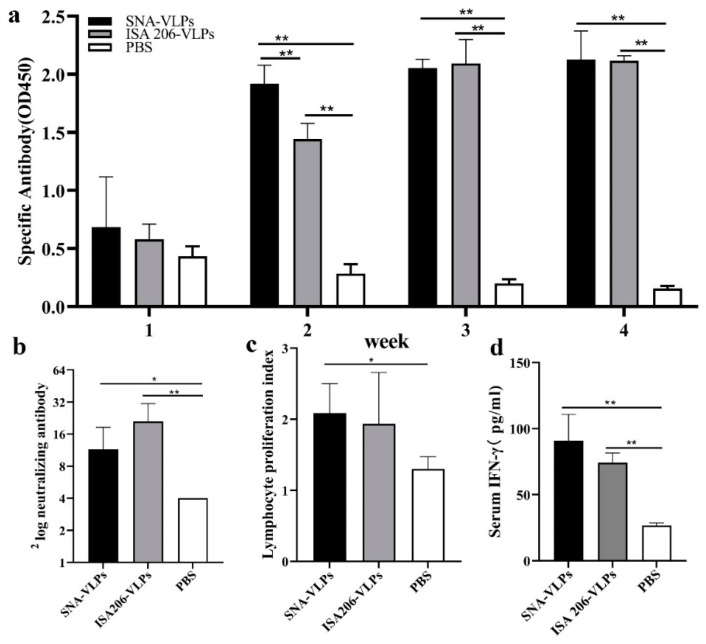
The results of the immunization experiments in guinea pigs. (**a**) The specific antibody levels in the guinea pigs immunized with various vaccines. (**b**) The levels of the neutralizing antibody. (**c**) The lymphocyte proliferation index. (**d**) The IFN-γ levels in serum. * *p* < 0.05, ** *p* < 0.01.

**Table 1 nanomaterials-12-03934-t001:** Protection of guinea pigs after challenge with FMDV.

Group	Guinea Pigs Number	Protection	Rate of Protection (%)
FMD VLPs-SNA	6	5	83.3% (5/6)
FMD VLPs-ISA206	6	5	83.3% (5/6)
PBS	6	0	0 (0/6)

Note: When neither the inoculated limb nor the uninoculated limb and vesicles were swollen, this was judged as protection; When both the inoculated limb and the uninoculated parts were infected, this was judged as unprotected.

## Data Availability

Not applicable.

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
