# Peer review of "Development and Efficacy Evaluation of a Novel Nano-Emulsion Adjuvant for a Foot-and-Mouth Disease Virus-like Particles Vaccine Based on Squalane"

_nanomaterials, 2022, doi:10.3390/nano12223934_

Round 1

Reviewer 1 Report

This manuscript studies the basic formulation of squalane nano-emulsion adjuvant(SNA) developed by a pseudo-ternary phase diagram. The  SNA was emulsified with FMD VLPs, and animal experiments were performed to evaluate the immune response of these VLPs.

Overall, this manuscript appears to be a difficult one to understand. It is desirable to revise the paper to make it easier for readers to understand. Please make corrections to complete the manuscript by giving your opinions on the following matters.

Main revision

1. Overall, I hope the manuscript is written in an easy-to-understand manner for the reader. especially,  Fig 1, 2.

2. There are many unscientific expressions. In terms of materials and methods, it is expressed by putting too easy a formula.

3. In the case of Figures 1 and 2, no matter how much you look at it, it is not easy to understand and the explanation is not friendly.

4. The case of Table 1 is also considered to be an unscientific table. It looks like a simple data package with a lot of information in it.

5. It would be good to see the correction of English by a native speaker.

Minor revision

1. It seems necessary to rectify the occasional typos (ex, SAN) and abbreviations.

2, There should be an explanation for the abbreviation SNA in the abstract.

3. In Figure 6, the descriptions of B and C should be changed.

Author Response

Dear reviewer:

   Mang thanks for the insightful comments and suggestions on our paper(nanomaterials-1947579). We have learned much from your comments, which are fair, encouraging and constructive. After carefully studying the comments and your advices, we used the "Track Chang" function when revising the manuscript.

Reviewer 2 Report

Shi et al. used FMD-VLPs to immunize mice and guinea pig to obtain protective immunity. They compared two different adjuvants to boost the vaccination.

Major comment: Overall, it seems that only minor differences were seen between the two adjuvants used. Throughout the text, the authors appear to claim SNA being superior to ISA, while there is only little evidence to back up this view. I would therefore recommend to revise the text significantly to provide more objective view.

Detailed comments:

1) When using bacterially expressed proteins as antigens, there is high risk for endotoxin contaminants. The endotoxins should be quantified and the measured levels reported.

2) Also, the concentration of DNA in the antigen sample should be reported.

3) Page 7: How did you detect bound antibodies after applying guinea pig serum on immobilized FMDV? The current statement "Dilutions of horseradish peroxidase-labeled guinea pig serum" does not make sense I think.

4) Section 3.1 should be revised to clearly indicate how the choice about reagent concentrations was made. Current description is not fluent enough to understand the key decisions.

5) Page 9: what was the method used to determine the PDI?

6) PDI = 0.4 is typically considered as acceptable. Reconsider the wording, now authors state "suggests good dispersion".

7) Figure 3A: authors should also show how the nanoemulsion appears without VLPs in TEM analysis. Also a sample containing VLPs only should be analyzed with TEM (as now shown in Figure 3b).

8) In Figure 5b, the Y axis ticks should be positioned so that 100% is easily detectable. Maybe 120, 100, 80...?

9) page 12 "animal organism" -> "enhance the immune response in BALB/c mice as compared to VLP alone"

10) Page 12: Authors state "SNA vaccine produced the highest IgG1 among the four experimental groups". According to Figure 6a, this seems not to be the case - no difference between SNA and ISA were observed.

11) Page 13: Authors claim that SNA-VLPs produced higher antibody levels than ISA-VLPs, but according to Figure 7a, there is no such difference observed. Instead, they seem to produce highly similar antibody response.

12) The difference in IgG1 and igG2 levels are also modest between adjuvanted groups.

13) Figure 6d: the difference between ISA and SNA appears not to be statistically significant. Please adjust the text accordingly. And please consider removing the last sentence in the first paragraph in Page 13.

14) Page 13-14: Authors state "Figure 7c showed the lyphocyte.... and the SNA-VLPs immunization group was higher than the ISA206-VLPs...."
According to statistical analysis, this statement is not true. A difference may be seen, but it is not statistically significant. Therefore such claim should not be made.

Round 2

Reviewer 1 Report

Several problems have been improved. 

But, there are errors with the misspelling. 

The typos should be improved

   (1) in Table, FMD VLPs-IA206 ->FMD VLPs-ISA206  

    (2) in page 15, SAN-VLP -> SNA-VLP

Reviewer 2 Report

Authors addressed the comments properly and the manuscript was improved. I still noticed the following issues:

1)           Abstract “More importantly, immunization with one dose of SNA-VLPs resulted in total protection from the FMDV challenge in guinea pigs, the protection rate was the same as the ISA206-VLPs, which reached 83.3%.” -> Rephrasing is needed, 83.3<100.

2)           Page 3: Twenn60  -> Tween60. Also check the spelling of the detergents throughout and harmonize.

3)           Page 5: The volume and amount (micrograms) of the SNA injected into mice tissue should be provided.

4)           Figures 1&2: “Relatic area” is unclear term and should be rephrased.

5)           The Table errorneously reports 5 out of 6 animals protected by PBS in virus challenge? Please doublecheck.
